# How Reliable Are Networks? A Bayesian Modeling Approach

## Abstract

Network analyses of white matter pathways linking brain regions—noninvasively extracted from diffusion magnetic resonance imaging—hold great clinical application promise. However, these networks display low reliability at the level of single brain connections, severely limiting inference. We present a Bayesian modeling framework to assess the reliability of network connections across repeated measurements. We model connection strength as a mixture of two probabilistic components: one representing the presence of a true connection, and its true absence. Using simulated, repeated-measures data, we estimate the posterior distribution of connection strengths and quantify the reliability by examining the spread of these distributions. The model was sensitive to connections with varying levels of reliability. However, it underestimated the probability that a connection is absent, and failed to recover the parameters after generating data with the same model.

## 1 Introduction

A wide range of conditions, including schizophrenia [Griffa et al., 2015] and bipolar disorder [Fernandes et al., 2019], are thought to arise from altered brain connectivity. However, network representations derived from diffusion MRI (dMRI) yield unreliable estimates [Maier-Hein et al., 2017, Thomas et al., 2014, Nakuci et al., 2023], hindering biomarker discovery and clinical translation. Post-processing methods such as streamline filtering [Smith et al., 2013] can improve robustness but do not quantify residual uncertainty.

Bayesian approaches offer a principled way to assess edge uncertainty [Hinne et al., 2013], and have been used to model disease-specific alterations [Peterson et al., 2020] and causal interactions [Dang et al., 2018]. Here, we validate a Bayesian framework to (1) quantify the reliability of fiber density estimates for each structural connection (SC), and (2) generate an atlas classifying connections as likely present or absent. This approach produces connectivity estimates with explicit posterior confidence for each edge, enabling more reliable interpretation of SC and, ultimately, improved clinical applicability.

## 2 Methods

### 2.1 Model specification

We model white matter track density $D$ as a mixture of two components: one for absent connections ($C = 0$) and one for present connections ($C = 1$), where $C$ is latent. Noise, motion, and processing variability can yield nonzero densities even for $C = 0$; this component is modeled with a fast-decaying exponential distribution. We model true connections ($C = 1$) to follow a normal distribution centered on the connection strength, truncated at zero since fiber densities are nonnegative. The prevalence of either component is modeled by the probability $\pi_0$ that the connection is truly absent:

$$P(D) = P(D|C = 0)P(C = 0) + P(D|C = 1)P(C = 1)$$
$$P(D) = \pi_0 \cdot \text{Exp}(\lambda) + (1 - \pi_0) \cdot \mathcal{N}^+(\mu, \sigma), \tag{1}$$

where $\text{Exp}(\lambda)$ is a decaying exponential distribution with a rate $\lambda$, and $\mathcal{N}^+(\mu, \sigma)$ is a normal distribution truncated at zero, with a mean $\mu$ and a standard deviation $\sigma$.

**Prior distributions.** We set $\pi_0 \sim \text{Beta}(2, 5)$ to model probabilities in [0,1], centering on the 5–40The rate $\lambda \sim \text{Gamma}(1, 10)$ ensures positivity with flexible deviation, and $\sigma \sim$ HalfNormal(0.6) reflects positive and small expected variability in fiber density. Because the real dataset we will fit the model on includes only 36 repeated diffusion MRI sessions, we fixed $\mu$ to the mean fiber density across repeats rather than estimating it.

**Model implementation.** We implement and fit the model using PyMC [Abril-Pla et al., 2023]. All experiments are conducted on an Intel(R) Core(TM) i9-10980XE CPU @ 3.00GHz, 36 cores, 62 GB of RAM, Ubuntu 20.04.6 LTS. For inference, we used the No-U-Turn Sampler (NUTS), PyMC's default Hamiltonian Monte Carlo algorithm, with 4 chains and 2000 posterior draws per chain following 1000 tuning steps.

## 2.2 Validation

To simulate a single subject scanned 36 times (data we will leverage to quantify within-scanner edge-wise reliability using the model), we repeated a reference SC matrix from the atlas Alemán-Gómez et al. [2022] (CC-BY-4.0 license) with different realizations of bi-modal noise. We added Gaussian noise $\mathcal{N}(0, 0.2)$ to all connections mimicking measurement noise and a stronger noise $\mathcal{N}(0, 0.5)$ to those below the $40^{\text{th}}$ fiber-density percentile ($\leq 4.76$) to reflect the empirical observation that weaker connections are less reliable. This simulation was used to validate our model through three experiments. Parameter estimates were visualized as heatmaps, with connection groups highlighted via transparency masks, and group differences tested using two-sample t-tests (`ttest_ind`, SciPy [Virtanen et al., 2020]). The model was fit assuming independent edges, using four PyMC chains per edge (`cores=1`) and up to 20 parallel `joblib` jobs (total runtime: 9 h 5 min; 107 s/edge).

**Experiment 1.** We evaluate whether the model could detect the varying levels of edge-wise reliability we injected in the simulated SC, with reliability quantified as the standard deviation $\sigma$ of the normal component in the estimated posterior distribution.

**Experiment 2.** We assess whether the estimated $\pi_0$ correctly identified connections consistently absent across all 66 subjects used to build the connectome atlas Alemán-Gómez et al. [2022] as truly absent and all other connections as truly present.

**Experiment 3.** We systematically assess the model's ability to recover known parameter values. We generate SC matrices by fixing the true parameter values in Equation (1) and sampling from the posterior distribution. For each configuration, we fit the Bayesian model to the simulated data 10-30 times and compute the relative root mean square error (RMSE) between the true values and the posterior means. The true parameter values used are listed in Equation (2):

$$\pi_0 = 0.1, \quad \lambda = 2.0, \quad \sigma = 0.5,$$
$$\mu \in [0.1, 0.2, 0.5, 0.7, 1.0, 2.0, 3.0, 5.0, 8.0, 20.0, 30.0, 1000, 10000] \tag{2}$$

Since fiber density best distinguishes true from false connections, we varied the mean connection strength $\mu$ while keeping other parameters fixed. We set $\pi_0 = 0.1$ because the model consistently estimated low $\pi_0$ across connections (Figure 2). The choices $\lambda = 2$ and $\sigma = 0.5$ match the noise characteristics from the simulated SC matrices. The $\mu$ values span the observed range of average streamline counts in the reference SC matrix. Each $\mu$ fit (30 repetitions) took about 7 min 40 s, totaling roughly 1 h 40 min for all values.

## 3 Results and Discussion

**The model is sensitive to connections with varying levels of reliability.** Experiment 1 shows that the estimated $\sigma$ is systematically higher for connections with lower fiber density—those to which we added more noise (Figure 1). The model identified the two latent $C$ groups ($p < 0.001$).

**The model underestimates the probability of absence.** Figure 2 shows that, as expected, $\pi_0$ is low for consistent connections (Panel B), but for truly absent ones (Panel A), it remains lower than expected (max = 0.16 instead of $\approx 1$).

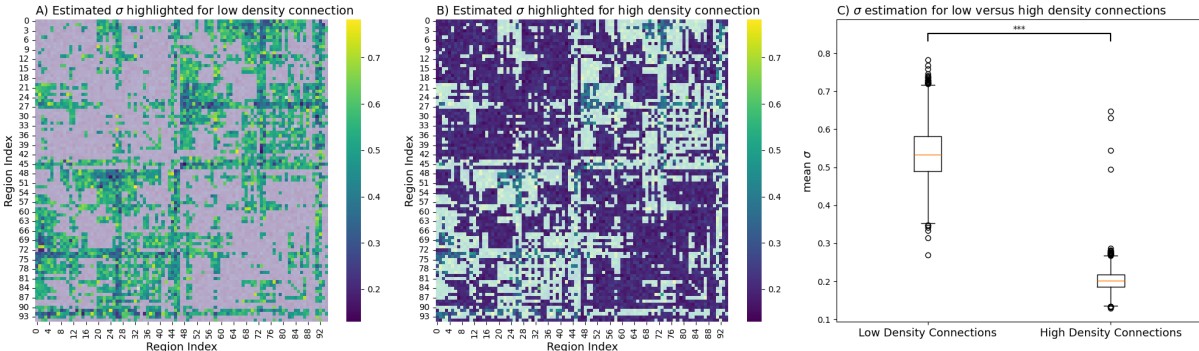

**Figure 1:** The model is sensitive to connections with varying levels of reliability.

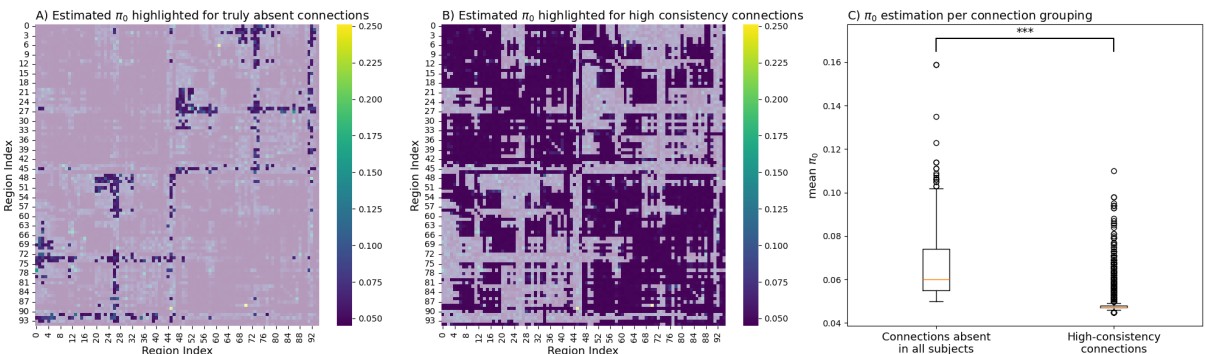

**Figure 2:** The model correctly predicts low $\pi_0$ for connections consistent across subjects, but underestimates $\pi_0$ when connections are truly absent.

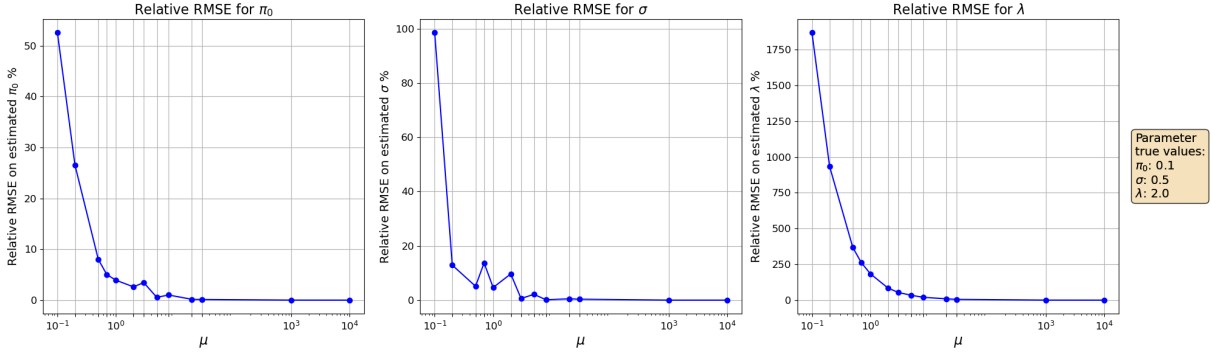

**Figure 3:** The model does not accurately estimate parameter values when the normal component of the posterior distribution is centered near zero.

**The model does not accurately estimate parameter values when the normal component of the posterior distribution is centered near zero.** Figure 3 shows that relative RMSE is high when $\mu$ is near zero, indicating poor parameter recovery, but estimation improves markedly as $\mu$ increases. To avoid misestimating $\pi_0$ for truly absent connections, this limitation should be addressed before applying the model to the real SC matrices.

## 4   Conclusion

This study establishes a foundation for embedding uncertainty into network representations, in particular, structural brain connectivity extracted with dMRI.

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
