# OpenReview forum: "How Reliable Are Networks? A Bayesian Modeling Approach"
_EurIPS.cc/2025/Workshop/MedEurIPS — EurIPS 2025 Workshop MedEurIPS Submission_

### Official Review · Reviewer_Eyri · 2025-10-29
**Bayesian Mixture Model for dMRI Network Reliability**

**Rating:** 7
**Confidence:** 4

**Review:**

`Overview:`
This paper addresses the critical issue of low reliability in dMRI-based structural connectivity networks, a known problem that severely limits their application for clinical biomarker discovery. The authors propose a Bayesian model to quantify the reliability of each network connection. The model smartly conceptualizes connection strength (fiber density) as arising from one of two latent states: a "truly absent" connection, modeled by an exponential distribution (to capture noise), or a "truly present" connection, modeled by a truncated normal distribution. *The model is validated on simulated repeated-measures data*.

`Interesting:`
- This work applies a probabilistic ML method to solve a foundational challenge in computational neuroscience and medical imaging.
- The novelty lies in moving beyond simple filtering to a formal Bayesian model that provides a posterior distribution for each connection, allowing for explicit uncertainty quantification.
- The authors clearly report both the model's success and its failures.
- The model is sensitive to varying levels of connection reliability (a positive result from Exp 1), but it fails to correctly estimate the probability of absence ($\pi_0$) and cannot recover its own parameters from simulated data, especially for weak connections (a negative result from Exp 2 & 3).

The authors identify the main limitation: *the model's failure in parameter recovery near zero must be fixed before applying it to real data*. The submission would start a discussion by including the authors' hypotheses on why this failure occurs:
-  Is it a fundamental identifiability problem between the exponential and the truncated normal components when the mean $\mu$ is low?
-  Or could alternative priors or a different distribution for the "absent" component resolve this?

`Conclusion:` This is a clear, sound, and self-critical paper on an important topic. The finding that this intuitive model formulation fails is, in itself, a valuable contribution. It has the potential to stimulate a productive discussion on the right way to model uncertainty in connectomics.

`Score:` 5 Accept

---

### Official Review · Reviewer_Ektu · 2025-10-31
**This paper proposes a novel Bayesian modeling framework to address the notoriously low reliability of structural connectivity estimates derived from Diffusion MRI (dMRI) tractography.**

**Rating:** 6
**Confidence:** 3

**Review:**

``Strengths``
1. The work introduces a statistically rigorous method to embed uncertainty into structural brain network representations.
2. The model successfully achieves its primary goal, demonstrating sensitivity to varying levels of reliability injected into the simulated data. Specifically, the estimated $\sigma$ was correctly higher for connections with lower fiber density/higher noise (Figure 1).
3. The paper tackles a fundamental limitation in connectomics (low inter- and intra-subject reliability), which is a major bottleneck for clinical translation and biomarker discovery.

``Weaknesses``
1. The model fundamentally underestimates the probability that a connection is truly absent ($\pi_{0}$), particularly for edges that are genuinely absent across subjects. The maximum estimated $\pi_{0}$ was only $0.16$ when it was expected to be near $1.0$ (Figure 2). This significantly limits the model's ability to create a reliable "presence/absence" atlas, a stated goal of the research.
2. Poor Parameter Recovery near Zero: The model exhibits high Relative Root Mean Square Error (RMSE) for the key parameters ($\pi_{0}$, $\sigma$, $\lambda$) when the mean connection strength ($\mu$) of the true connection component is close to zero (Figure 3).
3. The authors fix the parameter $\mu$ (mean fiber density for present connections) to the mean fiber density across repeats rather than estimating it. While justifiable given the small number of real-world repeated measures, this is a strong constraint that bypasses the need for the model to jointly estimate $\mu$ and $\sigma$, potentially masking further instabilities or poor mixing behavior in a truly generative setting.

---

### Decision · Program_Chairs · 2025-10-31

**Decision:**

Accept (Poster)

**Comment:**

Both reviewers find the paper conceptually strong and relevant, introducing a rigorous Bayesian approach to quantify uncertainty and reliability in structural brain networks.